# Development of Personalized Nutrition: Applications in Lactose Intolerance Diagnosis and Management

**DOI:** 10.3390/nu13051503

**Published:** 2021-04-29

**Authors:** Millie Porzi, Kathryn J. Burton-Pimentel, Barbara Walther, Guy Vergères

**Affiliations:** 1Laboratory of Human Nutrition, Department of Health Sciences and Technology, ETH Zurich, 8092 Zurich, Switzerland; millie.porzi@alumni.ethz.ch; 2Agroscope, Federal Department of Economic Affairs, Education and Research EAER, 3003 Bern, Switzerland; kathryn.pimentel@agroscope.admin.ch (K.J.B.-P.); barbara.walther@agroscope.admin.ch (B.W.)

**Keywords:** dairy products, epigenetics, functional foods, genetic testing, gut microbiota, lactase persistence, lactose intolerance, omics, personalized nutrition, polymorphism

## Abstract

Recent discoveries in the “omics” field and the growing focus on preventive health have opened new avenues for personalized nutrition (PN), which is becoming an important theme in the strategic plans of organizations that are active in healthcare, food, and nutrition research. PN holds great potential for individual health optimization, disease management, public health interventions, and product innovation. However, there are still multiple challenges to overcome before PN can be truly embraced by the public and healthcare stakeholders. The diagnosis and management of lactose intolerance (LI), a common condition with a strong inter-individual component, is explored as an interesting example for the potential role of these technologies and the challenges of PN. From the development of genetic and metabolomic LI diagnostic tests that can be carried out in the home, to advances in the understanding of LI pathology and individualized treatment optimization, PN in LI care has shown substantial progress. However, there are still many research gaps to address, including the understanding of epigenetic regulation of lactase expression and how lactose is metabolized by the gut microbiota, in order to achieve better LI detection and effective therapeutic interventions to reverse the potential health consequences of LI.

## 1. Introduction

Nutrition is an environmental variable of major importance for optimal health and disease prevention [1]. However, optimizing nutrition for health is challenging due to the highly variable individual response to diet [2], which results from the combination of internal factors such as a person’s genetics and microbiome, as well as external factors like stress and physical activity [1,3]. Recent advances in high-throughput “omics” technologies and bioinformatics tools have helped to better understand the inter-individual variation in response to food intake [1,3].

The notion of an individual response to diet is a central element of personalized nutrition (PN), also often referred to as “precision nutrition” or “individualized nutrition”, which can be defined as an approach that uses individual information, more recently based on “big data”, to develop customized nutritional advice, products, and services [4]. The overall goal of PN is to prevent, manage or treat diseases, optimize health and well-being using clinical assessments, genetic information, biomarkers, and any other relevant information about individuals [3]. The diagnosis and management of lactose intolerance (LI), a common condition with a strong inter-individual component, offers an interesting example for the potential and the challenges of PN.

The typical gastrointestinal symptoms of LI are specifically due to the maldigestion of lactose resulting from a lack of the lactase enzyme [5]. The normal physiological decline in lactase activity during early childhood leads to the appearance of LI in adults, the lactase non-persistence (LNP) phenotype, due to the inheritance of an autosomal recessive trait [6]. LNP is the ancestral type and most common phenotype associated with lactase gene expression worldwide, with a global prevalence estimated at 68% [7]. In contrast, lactase production into adulthood, the lactase persistence (LP) phenotype, is observed in the presence of a gain-of-function mutation and is inherited as an autosomal dominant trait [8,9]. The LP variant is not evenly distributed worldwide: high frequencies are observed in people from European descent and in populations with a long history of dairying activity [10]. The spread of farming during the Neolithic period correlates with the occurrence of the LP phenotype in human populations, with the earliest appearance estimated ~8000–9000 years ago in Europeans, ~2700–6800 years ago in African populations, and ~4000 years ago in Middle Eastern populations [11,12]. This evolutionary process provides anthropological evidence for gene-culture co-evolution, with a positive selection on the LP phenotype in relation to the domestication of dairying animals and consumption of their milk [13,14,15]. Indeed, LP would have conferred various selective advantages, including access to a major source of energy to improve nutritional status, but also a largely pathogen-free source of fluids to prevent dehydration in arid environments, and a valuable source of calcium for maintaining bone health [11,13,16]. 

The role of lactose metabolism in human evolution and nutrition is a fascinating case that brings together ancient evolutionary history with recent discoveries on lactase epigenetic regulation, as well as the influence of the gut microbiota on the LI state and novel PN approaches in LI care. In this context, this article aims to review the current approaches used for the diagnostic and management of LI, to investigate the impact of LI on human health focusing on the application of nutrigenomics tools, and to critically discuss how recent development in PN will impact diagnostic tools and therapeutic interventions for LI.

## 2. Lactose Intolerance: Current Clinical Management

### 2.1. Physiology and Pathophysiology of Lactose Intolerance

The digestion and metabolism of the disaccharide lactose, requires the hydrolysis of the glycosidic bond connecting its monosaccharides, galactose and glucose, by the enzyme lactase-phlorizin hydrolase, commonly known as lactase, which belongs to the beta-galactosidase family [17]. The lactase encoding gene (*LCT*) is localized on the long arm of chromosome 2 in position 21 (2q21). On the same chromosome, *MCM6* encodes the minichromosome maintenance complex component 6, a regulatory element that controls the expression of *LCT* [5,18]. Initially, lactase is synthesized as pre-pro-lactase, which contains a signal sequence that is then cleaved in the endoplasmic reticulum to form pro-lactase. During intracellular transport, pro-lactase becomes N- and O-glycosylated in the endoplasmic reticulum and Golgi apparatus leading to the mature form of lactase, which is then exported and anchored to the apical brush border membrane of the intestinal epithelial cells [19].

The lactase enzyme is abundantly present in the proximal part of the jejunum, while its presence progressively declines towards the ileum [20]. After hydrolysis, galactose and glucose sugars are actively absorbed across the intestinal epithelial cells and transported into the bloodstream to be used as a source of energy (Figure 1). When lactase is absent or deficient, unhydrolyzed lactose is able to reach the terminal ileum and subsequently enters the colon. An excess of undigested and therefore non-absorbable lactose will draw water from the bloodstream into the intestinal lumen via an osmotic effect, causing loose stools or watery diarrhea [17,21]. Within the large intestine, the undigested lactose is then cleaved into monosaccharides by the colonic microbiota. This bacterial fermentation process forms short-chain fatty acids (SCFAs), such as acetate, propionate, and butyrate. SCFAs can either be used by the intestinal epithelial cells or excreted in the feces. In addition, gases are produced from the bacterial fermentation of lactose, primarily hydrogen (H_2_), carbon dioxide (CO_2_), and methane (CH_4_), which increase intracolonic pressure. All these factors lead to gastrointestinal symptoms including flatulence, bloating, abdominal pain, cramps, and nausea [22,23]. However, the severity of the symptoms after lactose ingestion depends on the amount of lactose ingested, intestinal transit time, lactase expression, variability of intestinal microbiota, individual sensitivity, and psychological factors [21,24].

Currently, the exact prevalence of LI remains unknown, but it is acknowledged that it varies considerably among different ethnic populations [21]. LP and LNP are determined by assessing genetic polymorphisms associated with LP. At least five single nucleotide polymorphisms (SNPs) occurring upstream of the *LCT* gene have been associated with LP into adulthood: LCT-13910C/T and LCT-22018G/A in populations of European descent, LCT-13915T/G in African and Middle Eastern populations, LCT-14010G/C and LCT-13907C/G in some African tribes [10,11,25,26,27]. For the most common SNPs associated with LP in the European population (LCT-13910C/T and LCT-22018G/A), homozygotes (TT and AA) and heterozygotes (CT and GA) are indicative of LP, whereas the wild type (CC and GG) results in LNP [28,29]. In consequence, homozygous carriers of LCT-13910C/C and LCT-22018G/G are typically found to develop LI. Interestingly, heterozygous carriers of LCT-13910C/T and LCT-22018G/A have shown intermediate enzymatic activity with the occurrence of LI symptoms during situations of stress or with intestinal infections [30,31]. The currently known SNPs only provide a partial description of the mechanisms involved in the development of LI and the exact process by which the lactase is downregulated in early childhood still remains unclear [25,28,32,33].

### 2.2. Types of Lactose Intolerance

The complete or partial inability to digest lactose often leads to the same clinical manifestations of LI, though the causes of lactose maldigestion differ. LI refers to a syndrome in which the lactose maldigestion causes an onset of gastrointestinal symptoms including diarrhea, bloating, flatulence, nausea, abdominal pain, and cramps [34] (Table 1). It should not be confused with milk or dairy allergies that are characterized by an abnormal immunologic response that can develop into severe life-threatening anaphylaxis [35]. LI is typically caused by lactase deficiency (LD) which implies a reduced or absent lactase enzyme activity in the small intestinal mucosa. There are three main forms of lactase deficiency: congenital, primary, and secondary. Congenital lactase deficiency (alactasia) is a rare autosomal recessive pediatric disorder associated with an absence of lactase expression in newborns. Primary lactase deficiency (adult-onset hypolactasia) is the condition resulting from the progressive and physiological decline of lactase enzyme activity that typically occurs during childhood. Conversely, secondary lactase deficiency (acquired LI) is induced by small intestine disease or injury such as gastroenteritis, celiac disease, inflammatory bowel disease, chemotherapy, and antibiotics treatment [21,34]. Of note, ethnicity has been shown to be a more important determinant of susceptibility to developing lactase deficiency in many patients with inflammatory bowel disease rather than disease markers (particularly for cases of ulcerative colitis and Crohn’s disease that does not involve the small intestine, distal obstruction, or bacterial overgrowth) [36,37]. Generally, the non-genetic aetiologies of LI can be reversed if the cause can be eliminated.

### 2.3. Diagnosis of Lactose Intolerance

The diagnosis of LI relies in part upon the development of gastrointestinal symptoms resulting from lactose ingestion, not all of which can be assessed objectively and many which overlap with other conditions, notably irritable bowel syndrome in which lactose maldigestion may be accompanied by sensitivity to other fermentable carbohydrates described as FODMAPs (fermentable oligosaccharide, disaccharide, monosaccharide, and polyols) [5,38,39,40]. In practice, the diagnosis of LI is usually made on the basis of clinical suspicion supported by the positive response to a dietary challenge such as a trial period of a lactose-free diet [41]. However, several clinical diagnostic tests are available including blood, breath, and genetic tests, which can confirm a diagnosis even in the absence of gastrointestinal symptoms.

Diagnosis of LI confirmed by a jejunal biopsy for the in vitro assessment of lactase activity [42], is regarded as the “gold standard” test for LI diagnosis and can be used to exclude other gastrointestinal conditions. False positives are rare for this method but false negatives may occur due to irregular dissemination of lactase in the intestine. However, this method is almost exclusively used in clinical research because of its high cost, invasiveness, and need for highly specialized equipment [5,21]. In contrast, the oral lactose tolerance test (LTT) is a minimally invasive metabolic test, which consists in the determination of blood glucose levels at various times, following the administration of an oral overload of lactose (25–50 g). As the digestion of lactose results in an elevation of serum glucose, lactose malabsorption is indicated by a failure in blood glucose level to rise ≥ 1.1 mmol/L above the basal value, 60–120 min after lactose ingestion [17,21]. However, due to individual variability in the gastrointestinal transit time and glucose metabolism, false-positive and false-negative test results are relatively frequent for the LTT, reducing its use in clinical practice [43]. Alternatively, the lactose hydrogen breath test (HBT) has been widely adopted for the detection of an increase in H_2_ in expired air at several time points, after an oral lactose challenge (25–50 g). H_2_ is produced due to bacterial fermentation of non-digested lactose in the colon and can be indirectly assessed in breath (Figure 1). A rise of exhaled H_2_ of ≥20 ppm from baseline within 90 min after lactose ingestion is indicative of lactose malabsorption [44,45]. Even though this test accuracy is influenced by the gut microbiome, relatively high sensitivity and specificity have been reported for HBT, making it the most common type of LI test used today [21,43,46,47].

The gaxilose test is a more recent and non-invasive diagnostic test based on oral administration of 4-galactosylxylose (gaxilose), a synthetic disaccharide and structural analogue of lactose. The intestinal lactase hydrolyzes the gaxilose compound into D-xylose, which is then absorbed into the blood and subsequently excreted in the urine. The D-xylose levels in urine or serum can be quantified by colorimetric methods [48]. Besides its high sensitivity and specificity, the gaxilose test is easy to use, does not require specialized equipment, and only induces minimal subject discomfort [49]. Genetic testing has also emerged as a less invasive tool for supporting the diagnosis of LI and tests based on the most common SNPs that are linked to LP in the Caucasian population (LCT-13910C/T and LCT-22018G/A) have been developed [29]. However, the use of these SNPs cannot be applied as a global diagnostic tool, as other polymorphisms that confer LP have been identified in several African and Arabian populations [5,10]. Moreover, this method may not detect all SNPs associated with LI that exist within multi-ethnic populations [10,33,50]. Currently genetic tests have a limited role in diagnosing LI in the clinical setting, as none of them achieve perfect sensitivity and specificity and the results do not always correlate with clinical symptoms [5].

### 2.4. Management of Lactose Intolerance

The main treatment options for LI consists in preventing gastrointestinal symptoms by reducing or eliminating the amount of lactose in the diet or by taking oral enzyme replacement therapy. In order to manage their symptoms, people with LI should avoid eating high-lactose foods, such as fresh milk or cream [51], while ensuring an adequate intake of nutrients from other foods [52]. It is also recommended that individuals with LI eat lactose-containing foods together with other foods and that they favor small repeated intakes of lactose over one single meal with a high amount of lactose [53]. Fermented dairy products like hard cheese, quark, or yogurt are suitable for the majority of individuals with LI [54]. In fact, most aged hard cheeses naturally contain very little, if any lactose [51]. Yogurt usually still contains an appreciable amount of lactose, but delivers lactic acid bacteria with beta-galactosidase activity known to improve lactose digestion [51,52,55].

It should also be noted that lactose is a common additive in many processed foods, such as frozen meals, sweets, cakes, and sauces [53]. This so-called “hidden lactose” is used for its texture and flavor enhancing properties. Moreover, lactose is commonly used in the pharmaceutical industry as an excipient for oral medications [56]. The dose of lactose in oral solid-dosage form is generally small as most pharmaceuticals provide less than 2 g of lactose per day. Nevertheless, alternative medications might be necessary for individuals suffering from severe LI [57]. Overall, individuals with LI should be aware of these hidden or added sources of lactose and their possible impact on LI symptoms when combined.

To support the management of LI, alternative foods that are naturally lactose-free, such as soy products (e.g., tofu, edamame) and plant-based drinks (e.g., soy, almond, and rice milk) are often recommended as an alternative for high-lactose foods [58]. In addition, the food industry has developed many “low-lactose” and “lactose-free” products using diverse processes to remove lactose from lactose-rich dairy foods [52]. Lactose can be physically removed from milk using ultrafiltration or chromatographic separation followed by subsequent hydrolysis of the remaining lactose [59]. Thus, the sensory properties of lactose-free milk produced are not affected, but this process may also remove some valuable minerals, like calcium [60]. Alternatively, lactose-free milk can be obtained by enzymatic hydrolysis of lactose to its monosaccharides, glucose and galactose, using microbial beta-galactosidase [59]. This process is known to generate extra sweetness and may also have an impact on the nutritional value of the hydrolyzed milk [52]. More recently, the use of “A2 milk” has been shown to reduce some of the gastrointestinal discomfort associated with drinking ordinary cow’s milk in individuals with LI [61]. “A2 milk” is a variety of cow’s milk containing mostly A2 beta-casein (like human milk, sheep and goat’s milk) that does not metabolize to the peptide beta-casomorphin-7 (BMC-7), which is implicated in adverse gastrointestinal effects, including inflammation [62]. Interestingly, the gastrointestinal symptoms resulting from the ingestion of cow’s milk in LI individuals are decreased when milk enriched in A2 beta-casein is consumed in place of regular milk (which contains A1 beta-casein) [61]. This indicates that the gastrointestinal symptoms due to LI might be confounded by the beta-casein variant present in milk.

Finally, in the case where food-based approaches to manage LI are insufficient or not feasible, lactase enzyme replacements may be taken prior to consuming a lactose-containing meal [63]. This therapeutic option uses exogenous lactase for lactose digestion, like the fungal beta-galactosidase tilactase, which may be helpful as an alternative to dietary restriction and thus avoid possible nutritional deficiencies [64,65].

## 3. Prolonged Health Implications of Lactose Intolerance

Many studies have been conducted regarding the pathogenesis, diagnostic, and treatment of LI, but there is limited information on the long-term health consequences related to this dietary intolerance. Health effects of LI are mostly indirect effects related to lactose and dairy products avoidance. Indeed, many individuals with true or perceived LI may choose to withdraw dairy products from their diet, and this may put them at risk of nutrient deficiencies or other health problems. In addition, individuals with self-reported LI often restrict their intake of other food items, especially legumes and dried fruits, due to general concern about bloating and flatulence [66]. Over time, multiple dietary exclusions may put these individuals at risk of developing malnutrition [67].

The exclusion of lactose can have direct consequences on the absorption of other nutrients. For example, lactose enhances the absorption of minerals such as calcium, magnesium, and manganese, which are crucial for bone formation and for energy metabolism [68,69]. Another important consideration is the health impact of foods used to replace lactose in the diet. One feature of lactose is its relatively low glycemic index that might have a beneficial effect on metabolic health [70,71]. Therefore, particular consideration should be given when replacing lactose by alternative carbohydrates, such as maltodextrin, which have substantially higher glycemic indices and may consequently impact the glycemic response of some individuals, depending on various person-specific factors [70,72,73], as well as potentially inducing effects on the microbiota and microbial metabolites [74]. In the last few years, maltodextrin has been increasingly used in lactose-free infant formulas, although the effects on human energy metabolism and gut microbiota modulation are still largely unknown [75,76].

### 3.1. Lactose Intolerance and Bone Health

Dairy products are a readily accessible source of nutrients that are important for promoting bone health, including proteins, vitamins such as vitamin A, B2, B5, B12, K2, and vitamin D, as well as minerals, notably calcium, potassium, magnesium, phosphorus, zinc, and iodine [52,77,78,79]. Deficient intake of calcium in LI individuals has been associated with lower bone mineral density [80,81], raising concern about the risk of osteoporosis and fractures later in life for this population [82,83,84]. Consumption of calcium-rich foods such as green vegetables with high calcium bioavailability (e.g., kale, broccoli), as well as calcium fortified foods (e.g., plant-based milk, breakfast cereals) or calcium supplementation (e.g., calcium formate) seem reasonable alternatives for LI individuals who avoid dairy products [79,85,86], although no study has yet systematically addressed the efficacy of such approaches on bone markers in this population [81]. Vitamin D, together with proteins, vitamin K2, potassium, and magnesium, is crucial for calcium absorption and bone mineralization [87,88,89]. Recent findings from a Canadian population-based study showed that LI individuals had lower serum levels of vitamin D, likely due to reduced dairy intake [90]. Of note, in the United States and Canada fortification of milk with vitamin D is mandatory, whereas in other countries like United Kingdom, Spain, and Australia, this type of fortification is not systematic [91]. Alternative dietary sources of vitamin D are therefore recommended for individuals with LI, in particular oily fish, eggs, liver, and fortified lactose-free dairy products [92].

### 3.2. Lactose Intolerance and Non-Communicable Chronic Diseases

Total dairy consumption has been associated with a reduced risk of cardiovascular diseases [93,94], but there is still controversy surrounding the optimal diet for cardiac health [95,96]. Several clinical and epidemiological studies have also suggested that total dairy consumption has a neutral or moderately beneficial effect on the incidence of type 2 diabetes and obesity, although the research evidence remains contradictory due to a great number of intervening factors [32,97,98,99,100]. Interestingly, LI individuals with lower milk and dairy food consumption have been reported to have decreased risk of lung, breast, and ovarian cancers, which may also depend on their specific food habits or other genetic and lifestyle factors [101,102]. Epidemiological findings have indicated a possible protective effect of total dairy products consumption against colorectal cancer [103,104]. In addition, Li et al. recently demonstrated an inhibitory role of *Streptococcus thermophilus* on colorectal carcinogenesis via the action of β-galactosidase using in vitro and murine models. These results would support a potential prophylactic use of fermented dairy that contains the bacterium in colorectal cancer prevention [105]. Nevertheless, the role of fermented and non-fermented dairy products consumption in cancer development remains controversial [99,106,107].

### 3.3. Lactose Intolerance and Colonic Health

In the context of the emerging role of the gut microbiota in health, the notion of “colonic adaptation” via regular lactose consumption of dairy products that implies a change in the composition and functionality of the intestinal microbiota [108,109,110], suggests that lactose may have prebiotic properties [111,112] and associated health qualities. Indeed, progressive and regular lactose ingestion in individuals with LNP may increase the concentration and fermentation capacity of colonic bacteria that are then able to ferment lactose without H_2_ production, possibly leading to symptomatic improvements in LI [20,22,113,114], though the broader impact of these changes on health are less well-defined.

## 4. Application of “Omics” Tools to Research on Lactose Intolerance

By facilitating the study of the complex interactions taking place between the cellular machinery and nutrients [9], the advances in bioinformatic tools and high-throughput “omics” technologies, in particular, genomics, epigenomics, metabolomics, and metagenomics, have contributed to a better understanding and management of LI.

### 4.1. Application of Genomics for Genetic Testing of Lactase Persistence

The field of nutrigenetics analyses polymorphism frequency in a given population and provides information related to the metabolic response to nutrients for each individual genotype [115]. Today, only a small proportion of genetic tests are carried out by healthcare practitioners who play an essential role in translating and integrating this information into personalized healthy eating advice for their patients [4,116]. The test for polymorphisms in regulatory sequences of the *LCT* gene is clinically available in Europe for two common polymorphisms, LCT-13910C/T and LCT-22018G/A, associated with LP in the European population [29,43]. However, the use of these SNPs cannot be applied as a global diagnostic tool, given that a total of twenty-three LP variants have currently been identified in the *MCM6* gene in distinct geographic regions and population groups [5,10,43,117]. Moreover, the available genetic test may not detect all SNPs associated with LI that exist within the increasingly multi-ethnic European populations [10,33,50]. Table 2 illustrates the variability in allele distribution of four major *MCM6* polymorphisms in selected countries and populations.

A growing number of specialized companies are offering the public access to genetic and nutrigenetic information, via direct-to-consumer genetic testing (DTC-GT), without need for an intermediary health professional [121]. DTC-GT is advertised as a new tool for consumers who wish to investigate their genetic susceptibility and for those in search of tailored nutritional recommendations [122]. These tests are sold online, directly to consumers, and for home use. Usually, the company selling the test provides a sample collection kit (saliva or cheek swab), the collected sample is then sent for analysis by post, and the results are finally delivered to the consumer by post or via an online account. Along with providing information on the risk of monogenic disorders such as LI, DTC-GT providers often market and sell personalized meal plans, dietary supplements, or exercise programs [122]. However, the interpretation of genetic data is complex and context-dependent, and there is concern about potential errors associated with misinterpretation of results, that in the case of LI could lead to dietary restrictions that may be unnecessary or incorrectly applied in the absence of tailored advice from a health professional [123,124].

Self-reported LI prevalence rates vary from 8 to 20% depending on the population group analyzed [66,125]. However, the correlation between self-reported LI and objective laboratory tests was found to be low [21,66]. Self-perceived LI symptoms usually drives consumers’ food choices and lactose avoidance [53]. Up to 40% of the companies providing DTC-GT propose LI in their catalogue, making it one of the most investigated nutrigenetic traits [121]. Nevertheless, for the majority of DTC-GT companies, the genetic variant being tested is either not mentioned or presented with ambiguous nomenclature [121]. Since the nutrigenetic DTC-GT is not yet specifically regulated, the validity of these tests have been questioned regularly in the literature [122,126]. Indeed, a recent study showed a high false positive rate (40%) in genetic results, mainly in cancer-related genes, analyzed by DTC-GT [127]. DTC-GT reports relative to LI are not easy to interpret and it is generally difficult to assess their scientific reliability [121,122]. The impact of LI false positive test results on consumer’s dietary perceptions and health decision-making could be far-reaching, with potential unnecessary avoidance of all dairy products and possible adverse nutritional outcomes without proper medical monitoring [125]. Therefore, it is crucial that DTC-GT companies provide accurate and accessible information about analytical sensitivity and specificity, as well as clearly stating which LCT gene variant has been tested to facilitate interpretation of the results and ensure consumer protection.

### 4.2. Epigenetics of Lactase Regulation over the Lifespan

Besides the genetic inheritance of *LCT* gene variants, some epigenetics processes are also involved in *LCT* regulation [117]. The progressive decline of lactase production throughout childhood and the development of LI into adult life in LNP individuals may be related to epigenetic aging [19]. Recent findings by Labrie et al. have shown that the accumulation of epigenetic modifications altering lactase expression, such as DNA methylation, in aging cells, is dependent on the genetic variants regulating expression of the *LCT* gene [128]. By definition, DNA methylation is the addition of a methyl group to the cytosine base of DNA and is usually associated with the repression of gene expression. Using an epigenome-wide approach, Leseva et al. recently identified a position in the *LCT* promoter at which methylation levels were associated with the genotype at -13910C/T, the persistence or non-persistence phenotype and lactase enzymatic activity, higher methylation being associated with lower lactase activity [129]. These authors concluded that combining this epigenetic information with genetic information increases the predictive power of the data for identifying LI. Interestingly, Labrie et al. showed that LNP individuals display accumulation of methylated cytosines in both the *MCM6* and *LCT* genes, which results in poor *LCT* expression and the development of LI symptoms with age, whereas LP individuals were not susceptible to this age-related epigenetic modification [128]. Further findings of the same research group also showed that the changes in DNA methylation at the *LCT* promoter were correlated with lactase-mRNA levels found in intestinal cells. In fact, they observed a significant decrease of lactase-mRNA with age in LNP but not in LP individuals. Finally, an additional epigenetic modification of histone proteins, known to alter the chromatin structure and therefore the DNA accessibility, was described by these researchers [128]. This histone modification may also contribute to *LCT* gene repression in LNP individuals. Collectively, these recent findings improve the understanding of the combination of genes and epigenetic factors in the development of LI over time.

Surprisingly, only limited information is available regarding the mechanisms that regulate the decrease in lactase activity associated with weaning. Studies in rats have shown that this process is due to a decrease in transcription levels of the gene coding for lactase-phlorizin hydrolase (LPH) [130] (the rodent enzyme that hydrolyzes lactose) and that this process is not affected by the termination of milk ingestion [131]. Rossi et al. also proposed a transcriptional regulation of lactase activity after weaning in humans and they provided evidence for additional regulation at the post-translational level though the precise mechanisms still need to be elucidated [132]. The recent breakthroughs in epigenetics thus pave the way for further research in the regulation of lactase activity during the weaning process in infants.

### 4.3. Metagenomics, Lactose Intolerance, and the Gut Microbiota

Advances in sequencing technologies have led to new insights on how dietary pattern can influence the population, structure, and dynamics of the gut microbiota [133]. It has also supported understanding of how the homeostatic equilibrium of the gut microbiota contributes to the host metabolic functions and health status [134], as well as its known role in offering resistance against colonization by exogenous pathogens [135]. On the one hand, the interactions between dietary components and the gut microbiota generate many health benefits for the host (symbiosis), including the ability to utilize lactose and other carbohydrates, which cannot be metabolized in the small intestine. On the other hand, imbalances in the composition and activity of the gut microbiota (dysbiosis) may differentially influence the susceptibility of individuals to develop chronic metabolic diseases or immune-related disorders, as well as contributing to the generation of LI symptoms [136]. In LNP individuals, non-digested lactose may be fermented by commensal colonic bacteria to different metabolites including SCFAs, which can be used as an energy source for intestinal epithelial cells and enhance the gut epithelial barrier function [136]. A low carbohydrate fermenting capacity of the colon is implicated with the occurrence of diarrhea [137,138], while a larger number of colonic bacteria, i.e., a higher lactose fermentative capacity, may contribute to the reduction of LI symptoms [139].

Recent findings have shown that prebiotics, such as galacto-oligosaccharides (GOS), could contribute to alleviating LI symptoms by acting as a substrate for specific commensal colonic bacteria, especially *Lactobacillus* and *Bifidobacterium* populations, that are capable of fermenting GOS and lactose [140,141]. The stimulation of the growth and activity of lactose-metabolizing colonic bacteria by GOS has been found to increase the fermentation of lactose into glucose, galactose, and SCFAs [142]. There is some evidence that this “colonic adaptation” reduces gas production in the large intestine and could potentially improve lactose digestion and tolerance [22,140,141,143]. The developments in functional metagenomics have also helped to better understand the functionality and mode of action of probiotics. The latter are live microorganisms, typically bacteria and yeasts, that confer specific health benefits and are considered as nutritional therapeutic tools to promote the balance of gut microbiota, improve digestion, and support the immune system [144]. Certain probiotic strains, such as *Lactobacillus acidophilus* (used in yogurt), present a beta-galactosidase hydrolytic activity, which enhances intestinal lactose metabolism and may consequently alleviate LI symptoms [65,145]. Additionally, a recent study has found that the ingestion of a combination of different probiotic strains with beta-galactosidase activity induces more effective intestinal lactose digestion, helping to reduce LI symptoms [146]. Future approaches could include the development of “designer” symbiotic products containing both probiotics and prebiotics selected to improve lactose digestion based on the individual gut microbiota signatures [22]. In such synbiotics, the prebiotic compound could selectively improve the survival of the probiotic organism and promote the growth of commensal bacteria required for a healthy gut microbiota.

### 4.4. Current and Future Perspectives of Nutritional Metabolomics

The recent progress in metabolomics tools have enabled nutritional studies to exploit the method to identify dietary biomarkers and metabolites present in biological fluids following a dietary intervention [147]. With time, metabolomics may become a standard approach to measure dietary exposure, as it is a more reliable and objective method than traditional tools, such as food diaries [148]. Metabolic profiling can also offer insights on the metabolism of dietary components to, ultimately, improve understanding of the relationship between diet and health. Metabolomics has recently been used to measure serum lactose concentrations following dairy intake [149]; although the intestine does not contain transporters for disaccharides, traces of lactose could be measured postprandially in human blood and urine, allowing more thorough investigation of lactose metabolism and nutrient transport processes. In combination with nutrigenetics, metabolomics can additionally provide valuable information about individual dietary responses in relation to specific polymorphisms [150,151]. Indeed, in our group, an untargeted metabolomics analytical strategy in the postprandial serum and urine samples of healthy men having ingested dairy products supported the identification of galactitol and galactonate, two metabolites produced in increased amounts by hepatic galactose metabolism in response to lactose ingestion in LP but not LNP individuals [152]. This observation led to the proposition that these metabolites could be used in the development of novel non-invasive lactose digestion tests to help screen for LI.

Overall, metabolomics offers a unique opportunity to understand the specific metabolic and health effects of diet on the individual, and as such is a powerful tool for the development of PN [153,154]. Further metabolomics research could help to determine the potential health benefits of both fermented dairy products that have known benefits in LI and novel lactose-free products [155].

## 5. Lactose Intolerance as a Research Model towards Personalized Nutrition

The recent advances in “omics” technologies, especially nutrigenomics, combined with the growing interest of consumers in health and wellness, are paving the road toward PN. The emerging field of PN is based on developing tailored nutritional recommendations to individuals or to groups of individuals who share the same phenotype, with the goal of promoting health and preventing diseases. Such knowledge may contribute to reduce the incidence of chronic disease, increase life expectancy, and also participate in the decrease of healthcare cost burden [3,156]. Applying nutrigenomics research to LI in diverse population groups holds potential to help identify sub-populations and groups with high prevalence of LNP, which could then guide targeted modifications of existing food-based dietary guidelines to ensure adequate nutrient intake while accounting for dietary restrictions typical for the group.

### 5.1. Personalized Diets and Population Health

PN has the potential to inform and define national dietary guidelines to promote healthy eating at a population level [157,158]. Public health dietary guidelines are shaped by population-specific ‘Dietary Reference Values’ (DRVs), a set of estimates that collectively define the optimal daily energy and nutrient intakes for healthy populations [157]. The Recommended Dietary Allowance (RDA) is a DRV that provides the average daily level of intake sufficient to meet the nutrient requirements of nearly all (97–98%) healthy persons, and can be used for planning the diets of individuals [159]. Although, these values aim to capture the dietary requirements of the whole population group described, they have a limited ability to specifically address individual requirements that are modulated by genetic and environmental factors. Moreover, the translation of the nutrient requirements to food-based dietary advice may fail to offer adequate dietary sources of certain nutrients for groups of individuals who systematically exclude multiple foods in a given food group. One of the potential uses of PN for improving public health nutrition is to define and identify sub-populations with distinct dietary requirements. Several studies have already shown that LNP individuals have a lower intake of calcium compared to LP individuals [53], and indeed many national dietary guidelines only promote dairy foods as a source of dietary calcium [160]. With advances in diagnostic tools, the targeting of LNP populations with specific dietary guidance is increasingly a possibility. While people with LI may consume dairy products (with or without lactose), the relatively high prevalence of LI, together with the different dietary adaptation strategies used to manage the condition, warrants consideration of the promotion of alternative dietary calcium sources for this sub-population. PN research may thus contribute to the revision of existing food-based dietary guidelines to better meet the needs of each LI individual.

Depending on individual tolerance, different amounts of lactose can be consumed without adverse gastrointestinal symptoms [23]. Currently, lactose is considered together with starch and intrinsic sugars in DRVs with no specific recommendations for lactose intakes [161]. However, in view of the inter-individual variation in lactose doses that provoke adverse gastrointestinal consequences in LI groups, it could be interesting to explore the use of lactose DRVs that are specific for lactose intolerant populations such as a metric akin to the Upper Tolerable Intake Level (UL). The impact of lactose malabsorption on gastrointestinal symptoms is also confounded by other physiological processes as indicated by the relatively high rates of false positive and negative diagnostics tests on LI. Nutrigenomics research is expected to lead to a better understanding of the processes, ultimately delivering a more specific estimation of individual lactose-tolerance threshold for LI.

### 5.2. Personalized Nutrition for Health Optimization, and Disease Management

PN is starting to replace the conventional “one-size-fits-all” approach to offer public health interventions at an individual level, while in clinical dietetic practice, nutrigenomic data can be incorporated into both the clinical assessment process and evaluation of dietary interventions to respectively define and tailor the dietary treatment. Indeed, with the recent advances in “omics” technologies, it has become more feasible to investigate the mechanisms and determinants of how individual responses to diet vary. For illustration, the tailoring of dietary interventions by microbiota profiles is increasingly explored in PN. Research on lactose digestion indeed suggests that LI could be improved or even reversed by personalized dietary interventions that modulate the gut microbiota, such as progressive and regular consumption of lactose [111,112] or the ingestion of selected pre- and probiotics based on gut microbiota signatures [20]. Promising new strategies are studying the use of GOS and combinations of probiotics strains to adapt the gut microbiota composition and diversity in order to reverse LI, at least to some extent [141].

With the increasing focus of research on gene-diet interactions and the growing number of SNPs identified, an improved understanding and management of monogenic and polygenic disorders has become possible [162]. In fact, SNPs can modulate the individual metabolic response to specific dietary components and influence disease risk [9,163]. However, the recommendations for PN are currently more easily developed for individuals with monogenic conditions, caused by a mutation in a single gene and involving a single dietary exposure [9,116,164]. Conversely, in polygenic diet-related disorders such as obesity, type 2 diabetes, metabolic syndrome, or cancers, the combination of multiple SNPs and the complex interactions with environmental factors are not yet fully understood [165,166], thus complicating the translation process of genetic research into PN [116]. At first sight, LI is indeed considered a monogenic phenomena that can be managed through a lactose-free or lactose-reduced diet. However, this monogenic view can be challenged in light of the many uncharacterized polymorphisms impacting on *LCT* expression as well as of the contribution of epigenetic mechanisms and the gut microbiota to LI. Compared to more complex disorders, LI provides, however, a relatively simple model for researchers to implement PN.

Genetically-tailored dietary recommendations may have a positive impact on motivation and sustained eating behavioral change, as well as health outcomes [167]. Consumers are increasingly concerned with the ecological and ethical issues associated with foods of animal origin including dairy products [168,169]. Combined with the increased awareness of consumers for their individual dietary needs [170,171], these concerns have brought LI to the forefront of consumer discussions on dairy products, making it both crucial and yet complex for nutrition scientists to inform the public on PN.

### 5.3. Food Market, Product Innovation, and Affordability

Advances in nutrigenomics research have led the food industry to develop personalized foods, supplements, and meal plans customized to meet nutritional needs, dietary restrictions, and lifestyle choices (e.g., gluten-free and vegan food items) [172,173]. Recently, there has been an increase in consumers’ demand for “free-from” products and functional foods, that provide health benefits beyond their nutritional value, and this market is expected to expand even more in the coming years [172].

For the management of LI, specific “personalized” products are already available such as cow’s milk that is enriched with lactase enzyme and probiotic-supplemented milk which is an interesting alternative to hydrolyzed dairy milk for LI individuals [52]. Future applications of nutrigenomics and PN could include the distribution of LI DTC-GT in pharmacies, to offer more accessible and quick results for a variety of polymorphisms associated with LI. Artificial intelligence tools could also be used to integrate genetic information and diet-related reports in the form of “smartphone coaching apps” providing 24/7 tailor-made diet recommendations to support LI individuals.

There is concern about the limited accessibility of PN to socioeconomically disadvantaged individuals, which may further widen existing health inequalities [158,174], particularly in light of the relatively high cost of genetic testing and professional nutrition services, along with the lack of adequate insurance reimbursement [4]. As a consequence, PN technologies and services are so far mainly oriented towards wealthy individuals seeking to enhance their health [175] or athletic populations looking to maximize their performance [176]. LI is not life-threatening and, in the absence of severe symptoms, tests to identify this clinical syndrome are usually not covered by health insurances. The average price for LI DTC-GC ranges from $100 to $200 but can be significantly higher [121]. A lifestyle market for LI tests could thus develop that will only be available to healthy individuals or societies.

### 5.4. Challenges for Developing Personalized Nutrition

PN holds huge potential for the future of health care, wellness, and food industries, and nutrition services that are illustrated in its application to diagnose and manage LI. However, PN is still in its early stages and there are many challenges that the sector must overcome.

The first level of challenge resides in the optimization and harmonization of methodologies as well as validation of the results derived thereof. Specifically, further work is needed to validate genetic or epigenetic markers that can predict the risk of developing diet-related diseases [177,178], as well as biomarkers suitable for early detection of disease [178]. Additionally, despite the rapid progress in metabolomics, there is still lack of clinically validated, easily measurable and cost-effective metabolic biomarkers for accurate and objective assessment of individual response to dietary intake [179]. In the case of LI, the genetic and epigenetic phenomena leading to this condition have now been identified but validated testing remains limited to the most prevalent SNPs and does not yet include epigenetics markers. Furthermore, the result of the combined effects of genetic determinants and epigenetic modulation on LI remains to be understood. Urinary metabolic biomarkers of hepatic galactose metabolism, such as galactitol and galactonate, have been identified in this review for their promising use as a noninvasive lactose digestion test [152]. However, this indirect measurement of LNP is dependent on the lactose dose and type of substrate ingested [180]. Therefore, there is a need for standardized testing procedures as well as validated cut-off values specific to each type of substrate before this method can be extensively deployed for LI detection in community and clinical practice settings.

Since the vast majority of scientific evidence for PN is based on retrospective or observational studies [3], there is also a need for larger randomized controlled interventions of longer duration, assessing genotypic as well as phenotypic traits, to evaluate the effectiveness and utility of PN interventions. The use of alternative, novel study designs such as “n-of-1” or “single subject” clinical trials that integrate the very concept of individual response to interventions in their design, may also be important for demonstrating the benefits of PN interventions [181]. LI is characterized by a variety of genetic and epigenetic mechanisms inducing various physiological intestinal processes. As these processes are essentially induced by a single disaccharide (i.e., lactose), LI provides an interesting case-study for scientists to successfully demonstrate the validity of the “n-of-1” approach.

The complexity of “omics” data has limited the translation process of research findings into effective and practical PN recommendations [116]. The integration of genetic testing into everyday clinical practice as a part of preventive medicine requires the support of trained medical staff. Currently, healthcare professionals are lacking sufficient genetic nutritional education as well as continuous training in order to be able to provide an accurate and integrative interpretation of the results, together with reliable practical advice to the patients [4]. In addition to genetics, the recent findings on the role of epigenetics and the gut microbiota on LI, adds further complexity for practitioners with regard to the clarity of the message they can deliver to their patients and, consequently, their credibility, LI being no longer explainable by a single penetrant genetic polymorphism. The implementation of LI PN-driven interventions requires more accessibility and affordability to extend its awareness and use [175], which will ultimately enhance data on diverse populations throughout the world and increase the reliability of LI genetics-based nutrition recommendations. These tests should nonetheless demonstrate clinical utility before being embraced by public health authorities.

A further challenge in the translation of PN is the current lack of harmonized regulation of certain food labels, lactose-containing products being a typical example of how labeling can compromise the willingness of consumers to make informed decisions on which foods are more suitable for them and on available alternatives food options [53]. Indeed, there is no agreement so far on a specific “lactose-free” or “low-lactose” logo, nor on a precise cut-off value for establishing a “lactose-free” labeling policy, except for infant formula [53,182,183]. Consequently, it is necessary to reinforce health claims legislation and food law regulation for such products in order to achieve adequate consumer protection and maintain personal freedom of choice [184,185].

Finally, an important challenge in PN implies ethical considerations. The biological variation and cultural diversity between individuals are often not properly appreciated: lactose intolerance is seen as a deviation from normality (abnormal or deficient state) whereas the biology that enables continued milk consumption in adulthood is well-received and regarded as an advantage [186]. The psychological impact after LI diagnosis should not be underestimated, as individuals living with LI often report lower quality of life scores [187], higher anxiety, and might be at higher risk of developing depressive disorders [188,189]. Ultimately, tailored nutrition products marketed as improving health may negatively influence the role of food in cultural and social relationships, as it may blur the distinction between food and drugs [190]. The medicalization of food products and the exaggerated attention on a healthy lifestyle may thereby impact eating patterns and could even exacerbate the development of eating disorders [191]. Thus, the implementation of PN strategies should not neglect the wider implications of making diagnostic tools and personalized dietary advice readily available.

## 6. Conclusions

LI is a well-studied clinical syndrome but is often misdiagnosed. Recent advances in the understanding of the molecular biology underlying LNP provide a roadmap for research that will improve the diagnosis and management of LI. In particular, understanding the epigenetic regulation of *LCT* gene variants across the lifespan, especially during weaning and aging periods, will contribute to a better characterization of the pathophysiology of LI. Extending the panel of genetic variants beyond the commonly investigated *MCM6* polymorphisms will improve the performance of genetic tests for LI diagnosis, particularly in view of the increased human mobility across distinct geographic regions. The precision and sensitivity of the current LI diagnostic tests will further be improved by combining them with functional tests based on the metabolism of lactose. Finally, clarifying the contribution of the intestinal microbiota to lactose digestion in LI could support the development of promising therapeutic strategies based on pre- and probiotics. In order for these advances to translate into services that will profit the consumer, basic research on LI will need to be complemented by data validating the newly discovered biomarkers as well as by standardized testing procedures that can be used in community and clinical practice settings. Although associated with relative mild clinical consequences, the genetic, epigenetic, and metabolic processes characterizing LI provide a very interesting model for how to translate basic research on diet-related diseases into PN.

## Figures and Tables

**Figure 1 nutrients-13-01503-f001:**
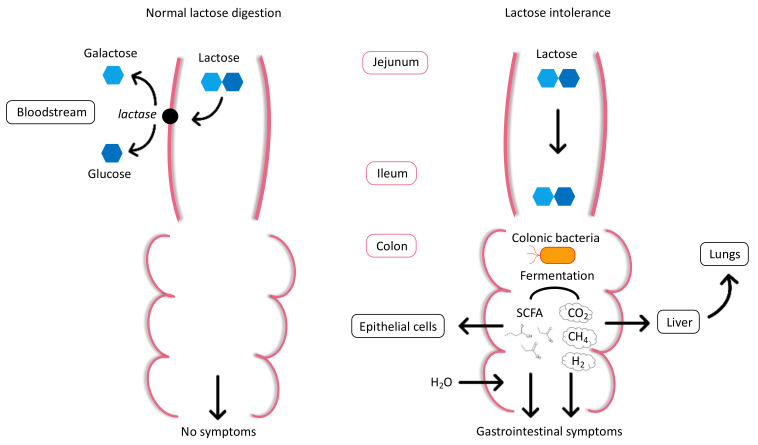
Normal lactose digestion and lactose intolerance.

**Table 1 nutrients-13-01503-t001:** Glossary of terms and definitions used to describe lactose digestion and metabolism [7,10,21,34].

Term	Definition
Lactose malabsorption	Failure to digest/absorb lactose due to primary or secondary lactase deficiency.
Lactose intolerance (LI)	Clinical syndrome in which the ingestion of lactose causes typical gastrointestinal symptoms such as diarrhea, bloating, flatulence, nausea, abdominal pain, cramps.
Self-reported LI	Individuals who perceive themselves as being LI without medical diagnosis.
Lactase deficiency	Lack or absence of intestinal lactase enzyme activity.
Congenital lactase deficiency	Rare genetic disorder in which lactase is already absent at birth.
Primary lactase deficiency	Progressive decline of lactase enzyme activity with age.
Secondary lactase deficiency	Reversible condition caused by illness or injury of the small intestine and resulting in deficiency of intestinal lactase enzyme activity.
Lactase non-persistence (LNP)	Most common phenotype associated with lactase gene expression worldwide. Characterized by lactase activity decline during early childhood.
Lactase persistence (LP)	Phenotype expressed by the continued activity of the lactase enzyme throughout adulthood.

**Table 2 nutrients-13-01503-t002:** Frequencies of various lactase persistence alleles in the *MCM6* gene in selected countries/populations.

Country or Population	Alleles	Frequency (%)	Reference
Sweden	LCT-13910C/T	73.7	[118]
Estonia	LCT-13910C/T	51.4	[119]
Saudi Arabia	LCT-13915T/G	59.4	[120]
Jordan	LCT-13915T/G	39.1	[26]
Tanzania	LCT-14010G/C	31.9	[11]
Kenya	LCT-14010G/C	27.6	[19]
Ethiopia (Afar)	LCT-13907C/G	20.0	[26]
Sudan (Afro-Asiatic Beja)	LCT-13907C/G	20.6	[11]

## Data Availability

Data sharing not applicable.

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
