# Peer review of "Development of Personalized Nutrition: Applications in Lactose Intolerance Diagnosis and Management"

_nutrients, 2021, doi:10.3390/nu13051503_

Round 1

Reviewer 1 Report

cvx

Manuscript ID: nutrients-1194901

The review article by Porzi M et al describes the concept of personalized (or precision) nutrition using lactose intolerance as an example where this type of intervention is valuable. The concept of personalized medicine also being applied to many chronic non – communicable diseases and therefore the paper is timely. Sections 1-3 review the basics of lactose digestion. Sections 4 and 5 discuss more recent findings of host food microbiome interaction and how the concept of PN can be modified. In section 5.4 the authors outline some features that need to be utilized to implement PN.Comments

The review is well written and focused on how to achieve the goal of personalized nutrition in lactose intolerance. Future studies, as the authors point out will require controlled trials, how specific treatments can be instituted at individual levels.

One difficulty with lactose intolerance is that in modern society the symptoms of LI are often merged with symptoms of functional gastrointestinal disorders. This is compounded by the public`s acceptance of LI as the primary culprit. Often patients with functional symptoms will already have tried lactose reduction before seeking medical advice.The authors discuss the point of self reported LI. However an important point is to clearly distinguish lactose malabsorptions status, based on testing from symptoms of intolerance. Here symptoms following indirect testing are not necessarily reliable for symptoms potentially related to daily living conditions. Some persons indeed have difficulties with consuming lactose while in others a number of other food sensitivities confer symptoms. There is a somewhat similar frequency of LI both in LP and LNP dominant populations. This notion is a “raison d’ètre” for the potential use of PN.

Minor comments

Pg 2, Line 63.  Vitamin D content of milk….. I think a comment should be inserted regarding source of vitamin D in milk since raw milk may not contain vitamin D (Mandrioli M et al Foods 2020, 9, 548; doi:10.3390/foods9050548).

Pg 4, Line 137. When considering inflammatory bowel diseases it is mainly Crohn’s disease involving the small bowel, or with distal obstruction and bacterial overgrowth that Lactose maldigestion may become a problem with addition of symptoms. In Ulcerative colitis, maldigestion of lactose follows ethnic distributions (Mishkin B et al Am J Gastroenterol. 1997;92: 1148-1153. Szilagyi A. Nutrition J 2016; 15: 67 DOI 10.1186/s12937-016-0183-8).

Pg 10, Line 431. FYI An interesting study was just recently published on the in vitro and in vivo (small animal) benefit of Streptococcus thermophilus in inhibiting colorectal carcinogenesis through variant β-galactosidase action (Li Q et al.Gastroenterology.2021, 160: 1179-1193). The study supports a potential benefit of fermented dairy products in prophylaxis against colorectal cancer. Human studies would need to confirm these findings.

Author Response

Comment

One difficulty with lactose intolerance is that in modern society the symptoms of LI are often merged with symptoms of functional gastrointestinal disorders. This is compounded by the public`s acceptance of LI as the primary culprit. Often patients with functional symptoms will already have tried lactose reduction before seeking medical advice.The authors discuss the point of self reported LI. However an important point is to clearly distinguish lactose malabsorptions status, based on testing from symptoms of intolerance. Here symptoms following indirect testing are not necessarily reliable for symptoms potentially related to daily living conditions. Some persons indeed have difficulties with consuming lactose while in others a number of other food sensitivities confer symptoms. There is a somewhat similar frequency of LI both in LP and LNP dominant populations. This notion is a “raison d’ètre” for the potential use of PN.

Response

We fully agree with Reviewer. The issue of misdiagnosis of lactose intolerance with other disorders is discussed in section 2.3 and we understand that Reviewer stresses its relevance in his comment without requesting that we further emphasize the issue.  

Minor comments

Comment

Pg 2, Line 63.  Vitamin D content of milk….. I think a comment should be inserted regarding source of vitamin D in milk since raw milk may not contain vitamin D (Mandrioli M et al Foods 2020, 9, 548; doi:10.3390/foods9050548).

Response

A discussion on vitamin D in ‘modern milk’ (i.e. raw milk, pasteurized milk, supplemented milk) does not fit to this paragraph, which describes the evolution of lactose intolerance in ancient time. We have therefore removed the reference to vitamin D in this sentence in line 64.

Comment

Pg 4, Line 137. When considering inflammatory bowel diseases it is mainly Crohn’s disease involving the small bowel, or with distal obstruction and bacterial overgrowth that Lactose maldigestion may become a problem with addition of symptoms. In Ulcerative colitis, maldigestion of lactose follows ethnic distributions (Mishkin B et al Am J Gastroenterol. 1997;92: 1148-1153. Szilagyi A. Nutrition J 2016; 15: 67 DOI 10.1186/s12937-016-0183-8).

Response:

We have taken this comment into consideration in lines 136-144 of section 2.2. The two references [36,37] proposed by the reviewer were accordingly added.

Comment

Pg 10, Line 431. FYI An interesting study was just recently published on the in vitro and in vivo (small animal) benefit of Streptococcus thermophilus in inhibiting colorectal carcinogenesis through variant β-galactosidase action (Li Q et al.Gastroenterology.2021, 160: 1179-1193). The study supports a potential benefit of fermented dairy products in prophylaxis against colorectal cancer. Human studies would need to confirm these findings.

Response:

This publication is now presented (lines 297-300, section 3.2) and referenced [105 ].

Reviewer 2 Report

Lactose intolerance (LI) is a common gastrointestinal condition worldwide. In their review, Porzi M et al. provide an excellent overview about the physiology, pathophysiology, diagnosis and management of LI.

This work has some merit because the authors elucidate also epigenetic aspects and provide suggestions of an individualized treatment optimization concept for LI. This part of the review differentiates from many other traditional reviews about LI.

The work is well written and is easy to understand for the readership. For the reasons mentioned this review should be published in “Nutrients”.

Author Response

No request for revision from reviewer 2